# Retrieval-Based Video Language Model for Efficient Long Video Question Answering

## Abstract

The remarkable natural language understanding, reasoning, and generation capabilities of large language models (LLMs) have made them attractive for application to video question answering (Video QA) tasks, utilizing video tokens as contextual input. However, employing LLMs for long video understanding presents significant challenges and remains under-explored. The extensive number of video tokens leads to considerable computational costs for LLMs while using aggregated tokens results in loss of vision details. Moreover, the presence of abundant question-irrelevant tokens introduces noise to the video QA process. To address these issues, we introduce a simple yet effective retrieval-based video language model (R-VLM) for efficient and interpretable long video QA. Specifically, given a question (query) and a long video, our model identifies and selects the most relevant $K$ video chunks and uses their associated visual tokens to serve as context for the LLM inference. This effectively reduces the number of video tokens, eliminates noise interference, and enhances system performance. Our experimental results validate the effectiveness of our framework for comprehending long videos. Furthermore, based on the retrieved chunks, our model is interpretable that provides the justifications on where we get the answers.

## 1 Introduction

With the rapid development of the Internet and the widespread use of cameras and smartphones, both individuals and businesses are generating massive amounts of video data every day in various fields such as entertainment, education, and technology. In such era of information explosion, understanding and extracting information from video content has become increasingly important to better meet people's needs and promote social progress. In this context, Video Question Answering (Video QA) emerges as an essential interface for extracting and delivering information from video content. Video QA systems allow users to ask natural language questions about videos and receive answers based on the visual (and auditory) information within the video.

There is a growing trend towards leveraging large language models (LLMs) for video QA (Maaz et al., 2023; Wang et al., 2023; Li et al., 2023b; Zhang et al., 2023b). On one hand, LLMs benefit from the vast knowledge acquired through training on enormous text corpora; on the other hand, they provide users with a more natural and intuitive way to interact with video data. Generally, vision tokens extracted from a video snippet are transformed and used as input (prompt) to the LLM, along with the text query. However, it is worth noting that the consumption of abundant vision tokens by a LLM can significantly increase the memory and computational burden, making it unaffordable for low-resource GPU agents. To mitigate this issue, Maaz et al. (2023) perform global spatial and temporal pooling on the video tokens, although this comes at the cost of losing detail due to pooling. Zhang et al. (2023b) aggregate video tokens using a Q-former with cross attention. Most of these methods are designed primarily for short-video QA tasks, where answer-related frames usually spread over the trimmed video snippet.

In practical application scenarios, users often pose flexible questions related to long videos (e.g., longer than 1 minute), in which the segments containing pertinent information for answering the questions constitute merely a small fraction of the entire video. The presence of answer-irrelevant tokens are redundancy and may potentially interfere with the video QA process, diminishing its

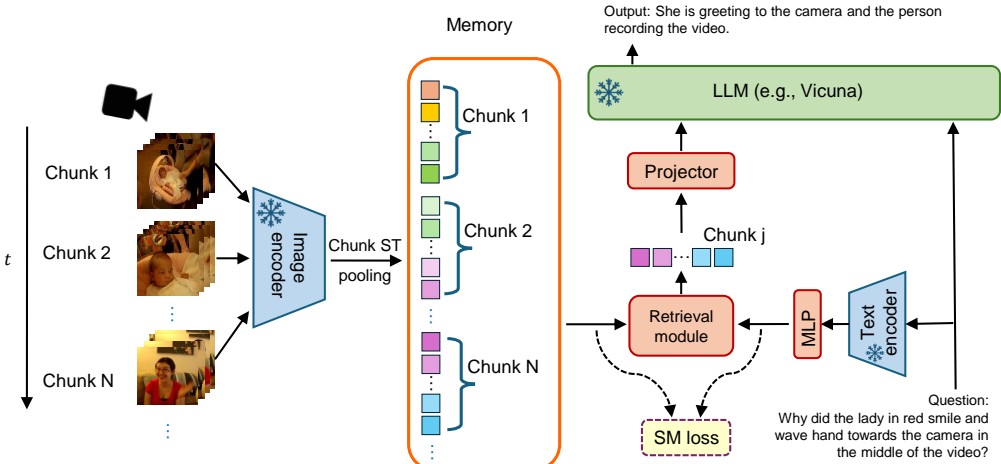

Figure 1: Illustration of our proposed retrieval-based video language model for efficient long video question answering. We encode an input long video into a sequence of video chunks, with each chunk represented by a set of spatial and temporal visual tokens. Question-guided retrieval is performed to find the top $K$ relevant video chunks, with their tokens as the context/prompt of the LLM for answer generation. Soft matching (SM) loss is used to regularize the retrieval related learning.

effectiveness. Therefore, it is imperative to develop a simple yet efficient framework that can handle long video QA tasks.

To address these challenges, we draw inspiration from biology and cognitive science. As we know, human working memory is a cognitive system responsible for temporarily holding and manipulating information necessary for complex tasks such as reasoning, learning, and comprehension (Thompson & Madigan, 2013). Faced with the vast amount of information stored in long-term memory, working memory selectively retrieves relevant information while filtering out irrelevant data for further cognition. Motivated by this, we aim to design a framework that is capable of identifying and concentrating on relevant video segments while filtering out irrelevant information, ensuring accurate and efficient question answering without imposing excessive computational demands.

In this paper, we propose a retrieval-based video-language model, R-VLM, for efficient and interpretable long-video question answering. Fig. 1 shows the overall framework. Specifically, given a long video and a question (query), we divide the video into a sequence of non-overlapping video chunks, with each chunk representing a short video segment, e.g., 4 seconds with a sample rate of 1fps. Note that we sample at a low frame rate in considering the memory limitation and video redundancy. To allow a chunk to contain dynamic temporal information, we use 4 seconds that are expected to contain temporal dynamics as the chunk unit. We then aggregate the encoded tokens within a chunk through spatial and temporal pooling to obtain chunk tokens, reducing some redundancy while preserving considerable local details. We perform question-guided retrieval to obtain the top $K$ most relevant chunks, and subsequently use the small set of tokens (after projection) from these chunks as input for the frozen LLM for video QA inference. In this way, we efficiently match and pass the most question-pertinent visual information to the LLM for inference.

Our framework demonstrates superior zero-shot generalization performance on several video QA datasets, outperforming the baseline method Video-ChatGPT (Maaz et al., 2023) by **6.8**%, **2.8**%, **4.8**%, **6.0**% in accuracy on the WildQA, QaEgo4D, lifeQA, and Social-IQ 2.0 dataset, respectively. Notably, the retrieved video chunks provide justification for the model's responses, offering interpretability to the video language model on where matters for answering the question.

Our contributions can be summarized below:

- We propose a retrieval-based video language model for efficient long video understanding. To the best of our knowledge, we are the first time to validate the feasibility of using retrieval for long video question answering with large video-language model.
- Thanks to the retrieval-based context/prompt chunk selection mechanism, our model is more interpretable, where the selected chunks provide insight on based on where and why the model generates the answer.

- Our method achieves significant performance improvement over the baseline method. The ablation studies demonstrate the superiority of our retrieval mechanism.

## 2 RELATED WORK

**Large Language Models.** LLMs (Radford et al., 2018; Brown et al., 2020; Ouyang et al., 2022) have experienced significant advances and demonstrated remarkable capabilities such as language understanding, reasoning, generation, and in context learning. These abilities enable LLMs to tackle diverse tasks with user prompts in a zero-shot fashion, reflecting their powerful adaptability and generalization. The Instruction tuning strategy as used in ChatGPT (OpenAI, 2023) improves the model's alignment with user intentions and optimizes the quality of generation. At the same time, many open-source instruction tuned models have emerged, such as LLaMA (Touvron et al., 2023), OPT (Zhang et al., 2022) and GML (Zeng et al., 2022), which have greatly promoted technological advancement and made significant contributions to the community.

**Vision-Language Models with Pre-trained LLMs.** Recent advances in LLMs have inspired and promoted the integration of pre-trained LLMs with visual processing components for multimodal understanding (Alayrac et al., 2022; Li et al., 2023a; Zhu et al., 2023; Liu et al., 2023; Huang et al., 2023). Flamingo (Alayrac et al., 2022) and BLIP-2 (Li et al., 2023a) are seminal works that leverage the pre-trained LLM and CLIP image encoder for zero-shot image-text abilities. BLIP2 (Li et al., 2023a) introduces a Q-Former to map the image tokens from the CLIP encoder (Radford et al., 2021) to the textual embedding space of LLMs. LLaVA (Liu et al., 2023) maps the image spatial tokens to the textual embedding space of LLMs using a linear projector. LLaVA (Liu et al., 2023), MiniGPT4 (Zhu et al., 2023), and mPLUG-owl (Ye et al., 2023) promote instruction following using image-instruction-following datasets.

Some works have focused on enabling video language understanding by integrating vision encoder and LLMs (Zhang et al., 2023a; Maaz et al., 2023; Li et al., 2023b; Wang et al., 2023). Video-LLaMA (Zhang et al., 2023a) uses a video Q-former to assemble the features from the pre-trained CLIP image encoder. Video-ChatGPT (Maaz et al., 2023) performs spatial and temporal pooling of the feature maps encoded by the pre-trained CLIP image encoder. The ensembled vision tokens are taken as input to the LLMs. Most of these works globally aggregate video tokens. They may work well for short videos. But for the long video language comprehension, they would be less efficient. The information of the question related video segments may be submerged to the global-wise token representations, making the reasoning difficult. We aim to design a simple yet efficient video language model for long video understanding. Chunk-wise representations and learnable retrieval are introduce to address the challenges.

## 3 PROPOSED RETRIEVAL-BASED VIDEO LANGUAGE MODEL

Given a lengthy video, it is time-consuming to watch the entire content to obtain the desired information. A Video QA system, which can automatically infer answers based on the long video and user's query (question), is in high demand. LLMs possess extensive world knowledge and reasoning capabilities, albeit at the expense of high computational complexity. Some previous works on multi-modal language models utilize a set of projected vision tokens as input (context) to LLM for inference (Li et al., 2023a; Zhu et al., 2023), where the inference cost is proportional to the number of input vision tokens.

Leveraging powerful LLMs to understand long videos presents challenges. Firstly, we expect to use only a small set of tokens as input to reduce computational costs. Secondly, as the video length increases, in general, there is a corresponding growth in the number of vision tokens and the overall amount of information. Representing a long video with very few tokens becomes difficult. Furthermore, question-irrelevant information may interfere with answer generation. Motivated by the retrieval mechanism of brain, we introduce a question-guided retrieval mechanism to identify and select a few relevant video chunks as context of LLM.

Fig. 1 illustrates the overall framework of our proposed retrieval-based video language model (R -VLM), designed for efficient long video question answering. The framework comprises several components: a frozen LLM, a frozen image encoder, a frozen text encoder, a memory for storing

video chunk tokens, a retrieval module for selecting the question-relevant chunks, a learnable MLP block, and a projector. Through end-to-end training with cross-entropy loss and the proposed soft matching (SM) loss, the MLP block is optimized to learn to identify the most relevant $K$ chunks, while the projector is optimized to align the selected vision tokens with the text (question) space. In the following subsections, we provide a detailed explanation of the key components and designs.

### 3.1 VIDEO TOKENIZATION

Video tokenization involves encoding the raw video into vision tokens, which will be processed and then passed to the LLM for inference.

*Chunking the video.* Given a long video sample $V_i \in \mathbb{R}^{T_i \times H \times W \times C}$ with $T_i$ frames, $C$ channels, height $H$, and width $W$ [1], we divide it into small non-overlapped video chunks (see Fig. 1, which are the basic units for question-guided retrieval. We set the duration of a chunk as 4 seconds, *i.e.*, $M = 4$ frames when frame rate is 1fps. We have $L_i = \lceil T_i/M \rceil$ chunks, where $\lceil \cdot \rceil$ denotes the ceiling function.

*Vision token extraction of a chunk.* The vision tokens of a chunk are obtained by encoding the images and performing spatial-temporal pooling. Following Alayrac et al. (2022) and Maaz et al. (2023), we adapt the pretrained language-image model CLIP (Radford et al., 2021) to extract per frame vision token features, which are suited for capturing high-level semantic information. For the $j^{th}$ video chunk $V_i^j \in \mathbb{R}^{M \times H \times W \times C}$, we obtain $M \times h \times w$ vision tokens extracted by the CLIP vision encoder, where $h = H/p, w = W/p$. $p$ denotes the patch size (which is 14 for ViT-L/14).

The preliminary token number of a chunk is large, *i.e.*, $M \times h \times w = 4 \times 16 \times 16 = 1024$. This would result in a high computational burden and large memory requirement for the LLM even though we only select a few chunks as input to LLM. We found that reducing the spatial resolution by a factor of 4 leads to marginal difference in performance while this can significantly reduce the number of tokens by 75%. Therefore, we perform spatial average pooling with stride 2 to have $M \times \bar{h} \times \bar{w} = 4 \times 8 \times 8 = 256$ tokens per chunk, where $\bar{h}=h/2$ and $\bar{w}=w/2$. This is equivalent to that we take the CLIP features of reduced resolution as the extracted feature. How to further reduce the number of tokens while preserving spatial-temporal features of a chunk? Motivated by Maaz et al. (2023), we perform spatial-temporal pooling on the token tensor. Particularly, global spatial average pooling for each frame is performed and thus we obtain $M$ tokens for the $M$ frames (*e.g.*, 4 tokens). Temporal pooling for each spatial position is performed to have $N= \bar{h} \times \bar{w} = 8 \times 8 = 64$ tokens. We have $N + M = 64 + 4 = 68$ tokens $F_i^j \in \mathbb{R}^{(N+M) \times D}$, with $D$ dimension for each token. Compared with the original 1024 in a chunk, the number of tokens has been reduced to 6.6%, which brings high calculation efficiency for the later LLM inference.

In contrast to Maaz et al. (2023), which performs global spatial and temporal pooling over the entire video, ours with chunks is capable of preserving local details and is better suited for long video QA.

### 3.2 QUESTION-GUIDED RETRIEVAL FOR CHUNK SELECTION

For a long video $V_i$ with abundant video chunks, we retrieve and select the $K$ most relevant chunks based on the question/query and use them as input to the LLM. The top $K$ retrieval aims to efficiently identify the most informative video segments for answering the given question, reducing the memory and computational burden to LLM and excluding the interference from irrelevant content.

We encode the question $Q_i$ into a feature vector $\mathbf{q}_i \in \mathbb{R}^D$ using the frozen CLIP text encoder $f_\theta$ and a learnable MLP block $\psi$ as

$$\mathbf{q}_i = \psi(f_\theta(Q_i)), \tag{1}$$

where the MLP block is a two-layer perceptron consisting of two fully connected layers and an ReLU activation function in between. This MLP transformation strengthens the correlation between the question representation and the potential corresponding chunk features for chunk selection. Achieving this is non-trivial, as we do not have any ground-truth locations of the question relevant chunks for supervision.

---

[1]Note that in considering the abundant redundancy in video, the video is already temporal down sampled to have the frame rate of one frame per second.

To identify the question-related chunks, we measure the affinity between the text feature vector and the chunk representation. For simplicity, we obtain the representation feature of the $j^{th}$ chunk for the matching by aggregating its $N + M$ vision tokens by averaging pooling as $\mathbf{v}_i^j = Avgpool(F_i^j)$. This is parameter-free, avoiding the need for large memory in optimization. We store the chunk representation features and chunk tokens in a memory to facilitate retrieval.

We compute the similarity scores between the question representation $\mathbf{q}_i$ and each video chunk representation $\mathbf{v}_i^j$ using cosine similarity metric as

$$s_i^j = \cos(\mathbf{q}_i, \mathbf{v}_i^j) = \frac{\mathbf{q}_i \cdot \mathbf{v}_i^j}{\|\mathbf{q}_i\|\|\mathbf{v}_i^j\|}. \tag{2}$$

We rank the video chunks based on their similarity scores $s_i^j$ (where $j = 1, \cdots, L_i$) and select the top $K$ most relevant chunks. The $K \times (N + M)$ vision tokens of these chunks are input to the LLM after a linear projection (a fully connected layer) on each token.

## 3.3 END-TO-END OPTIMIZATION

We train the network using an end-to-end optimization approach with the video instruction data. The image encoder, text encoder, and the LLM are frozen during the training. Only the parameters of the MLP block and projector are optimized. For a video-text pair, besides the cross-entropy loss between the generated prediction and groudtruth answer $\mathcal{L}_i^{pred}$, we introduce soft matching (SM) loss $\mathcal{L}_i^{\mathrm{SM}}$ to regularize the similarity learning as

$$\mathcal{L}_i^{\mathrm{SM}} = -\cos(\mathbf{q}_i, \bar{\mathbf{v}}_i), \quad \text{where } \bar{\mathbf{v}}_i = \frac{\sum_{j=1}^{L_i} e^{s_i^j} \mathbf{v}_i^j}{\sum_{j=1}^{L_i} e^{s_i^j}}. \tag{3}$$

Here $\bar{\mathbf{v}}_i$ is a weighted combination of the $\mathcal{L}_i$ chunk features, with weights determined by the similarities between the query and chunk features (see Eq.(2)). In order to maximize the cosine similarity score between query and the combined feature, the MLP block needs to be optimized to result in higher similarity to those question relevant chunks. Better optimized similarity scores lead to better selection of chunks to the LLM and thus superior performance.

The overall loss is as

$$\mathcal{L}_i = \mathcal{L}_i^{pred} + \lambda \mathcal{L}_i^{\mathrm{SM}}, \tag{4}$$

where $\lambda$ is a hyper-parameter that balances the contribution of the regularization term. We determine the value of $\lambda$ such that both loss terms are on the same order of magnitude. We set $\lambda = 10$ in our experiments.

# 4 EXPERIMENTS

## 4.1 IMPLEMENTATION DETAILS

Following Video-ChatGPT (Maaz et al., 2023), we use the Language-aligned Large Vision Assistant (LLaVA) (Liu et al., 2023) as our base model. We utilize the pre-trained CLIP ViT-L/14 (Radford et al., 2021) as our image encoder and extract the feature from the second-to-last layer as the $h \times w$ vision tokens of a frame. We use the fine-tuned Vicuna (7B) from LLaVA as our LLM. For the text encoder, we use the pre-trained CLIP ViT-L/14 text encoder and extract the class token feature of the penultimate layer. The number of neurons for the two fully connected layers of the MLP block is 1024 and 1024, respectively. The number of neurons for the projector is 4096.

We only fine-tune the MLP block and the projector while keeping the image encoder, the text encoder and the LLM frozen. We fine-tune the model for 3 epochs using video instruction data, with a learning rate of 2e-5 and a batch size of 40. Training our model takes about 24h on an A100 80GB GPU. We set $K$ to 5, resulting in $5 \times (64 + 4) = 340$ vision tokens as the input to the LLM. This is comparable to the number of vision tokens used in Video-ChatGPT (356 vision tokens).

Table 1: Information of the evaluation datasets, including the number of videos, the number of QA pairs, and the average video duration.

| Dataset | #Vid | #QA | Duration (sec.) |
|---|---|---|---|
| WildQA | 261 | 652 | 71.2 |
| QaEgo4D | 166 | 1854 | 495.1 |
| lifeQA | 49 | 372 | 74.0 |
| Social-IQ 2.0 | 144 | 876 | 60.0 |

## 4.2 DATASETS

We use Video Instruction Data collected by Maaz et al. (2023) for video instruction tuning. This dataset contains about 100k question and answer pairs based on the Activitynet dataset (average duration 180 seconds). The 100k QA pairs include various types of questions, including but not limited to describing the general content of the video, questions related to appearance, movement, trajectory, reasoning, and some tasks that require imagination.

We evaluate the generalization performance of our framework on four video datasets: WildQA (Castro et al., 2022), QaEgo4D (Bärmann & Waibel, 2022), lifeQA (Castro et al., 2020), and Social-IQ 2.0 (Wilf et al., 2023). The average duration of these datasets is larger than one minute. The average duration of QaEgo4D is more than eight minutes. Please see Table 1 for specific information about each dataset. Note that long video datasets are rare. Many popular video QA datasets are not suited for our study, where the average duration of videos are very short, *e.g.*, 15 seconds for MSRVTT-QA (Xu et al., 2017), 10 seconds for MSVD-QA (Xu et al., 2017), and 3 seconds for TGIF-QA (Jang et al., 2019). The collection and annotation of long video dataset is highly desired. We hope the research and industry committee work together to contribute large datasets of long videos to inspire more investigation in future.

## 4.3 EVALUATION METRICS

In this paper, we follow the metrics of accuracy and average score as proposed by Maaz et al. (2023) for performance evaluation, where we use ChatGPT (chatgpt35-turbo) to assist in judging the correctness of model predictions. ChatGPT accepts questions, groundtruth answers, and the model predictions as input. For each question answer pair, ChatGPT gives a binary judgment of "yes" or "no" to identify whether the predicted answer is correct or not, for accuracy evaluation. Moreover, a score of 0-5 is also given by ChatGPT to indicate how similar the prediction is to the answer. 0 represents the lowest score and 5 represents the highest score.

## 4.4 COMPARISON WITH OTHER VIDEO LANGUAGE MODELS

Table 2: Comparison with the other video-language models, including Video-LLaMA and the baseline method Video-ChatGPT. We report the accuracy (%)/ average score.

| Model | WildQA | QaEgo4D | lifeQA | Social-IQ 2.0 |
|---|---|---|---|---|
| Video-LLaMA (Zhang et al., 2023a) | 63.19/3.18 | **35.35**/1.94 | 35.75/2.32 | 55.78/2.90 |
| Video-ChatGPT (Maaz et al., 2023) | 58.00/3.30 | 29.74/2.43 | 33.87/2.55 | 57.73/3.26 |
| R-VLM (Ours) | **64.82/3.39** | 32.51/**2.45** | **38.71/2.61** | **63.65/3.40** |

We compare our final scheme *R-VLM* (Retrieval based video language model) with the baseline method *Video-ChatGPT* (Maaz et al., 2023) on four unseen datasets. We use the same training dataset as *Video-ChatGPT*. Both *Video-ChatGPT* and our *R-VLM* are built based on *LLaVA*, with similar model size. Table 2 shows the results. Our *R-VLM* outperforms *Video-ChatGPT* significantly by **6.8**%, **2.8**%, **4.8**%, **6.0**% in accuracy on the WildQA, QaEgo4D, lifeQA, and Social-IQ 2.0 dataset, respectively. Ours consistently achieves higher average score. Note that the number of vision tokens of our *R-VLM* is comparable to that of *Video-ChatGPT* (*i.e.*, 340 vs. 356), making a fair comparison. This demonstrates the effectiveness of our retrieval-based design. Our chunk-wise retrieval design facilitates the exploration of the most informative vision tokens and preservation of necessary vision details for QA.

We also evaluate the performance of other video language model, *e.g.*, *Video-LLaMA* (Zhang et al., 2023a), by testing their model directly. As shown in Table 2, our *R-VLM* achieves the best scores on all the four datasets. We found *Video-LLaMA* is prone to give detailed description of the entire video, whereas the answer is usually question-irrelevant. Fig. 2(a) shows some typical examples on the QaEgo4D dataset, where the answer includes much information while the desired answer is submerge in the overall answer. The accuracy metric is not perfect and trades such answer as correct. In contrast, our model provides more concise and accurate answers.

## 4.5 ABLATION STUDIES

Table 3: Comparison of different methods and chunk selection strategies. All these models are trained using the same video instruction data. *R-VLM* denotes our final scheme with learnable retrieval. *R-VLM w/ Uni.* denotes uniform sampling of $K$ chunks in our framework instead of retrieval-based sampling. *R-VLM w/ CLIP M.* denotes that we use the final CLIP class token feature of vision and text for matching in our framework, without learnable parameters for retrieval. We report the accuracy (%) / average score.

| Dataset | Video-ChatGPT | R-VLM w/ Uni. | R-VLM w/ CLIP M. | R-VLM |
|---|---|---|---|---|
| WildQA | 58.00/3.30 | 61.23/3.36 | 60.31/3.27 | **64.82/3.39** |
| QaEgo4D | 29.74/2.43 | 31.57/2.44 | 31.52/2.43 | **32.51/2.45** |
| lifeQA | 33.87/2.55 | 36.56/2.56 | 31.45/2.42 | **38.71/2.61** |
| Social-IQ 2.0 | 57.73/3.26 | 57.96/3.24 | 61.17/3.28 | **63.65/3.40** |

In this section, we study and validate the effectiveness of our retrieval designs, chunk-wise design and the soft matching loss, respectively.

**Effectiveness of Retrieval for Chunk Selection.** Under our framework, we compare our retrieval mechanism and the uniform sampling strategy for the selection of $K$=5 chunks as context input to the LLM. For the uniform sampling setting which we name as *R-VLM w/ Uni.*, the model selects $K$ from $N$ video chunks uniformly instead of based on question guided retrieval. Table 3 shows that our final scheme *R-VLM* with learnable retrieval-based strategy outperforms *R-VLM w/ Uni.* by 3.6%, 0.9%, 2.2%, and 5.7% on WildQA, QaEgo4D, lifeQA, and Social-IQ 2.0, respectively. For a long video, the video chunks relevant to the question usually account for a small portion of the entire video. It is difficult to hit these question-related chunks with uniform sampling. A large language model may can not correctly answer a question when it accepts video chunks not correlated with the question. In contrast, our learnable retrieval learns to select the chunks most relevant to the question, providing more reliable and less redundancy information to the LLM for effective inference. Fig. 2 which visualizes the selected chunks also validate this. Our retrieval based model is interpretable, providing where the model is based on to get the answer.

**Learnable Retrieval vs. Off-the-shelf CLIP based Retrieval.** The CLIP image encoder and text encoder are pre-trained to achieve vision and text caption alignment through contrastive learning over the last layer class token features (Radford et al., 2021). One may wonder how about the performance when we use the CLIP class token features for question and chunk feature matching under our framework. To answer this question, we design a scheme *R-VLM w/ CLIP M.* for comparison. For a chunk, we use the averaged class token feature of the last layer of the CLIP image encoder (CLIP ViT-L/14) as the vision chunk feature, and the class token feature of the last layer of the CLIP text encoder as the question feature for matching. There is no learnable parameters for the retrieval. From Table 3, we can see that our *R-VLM* with learnable retrieval consistently outperforms *R-VLM w/ CLIP M.*, upto 7.3% in accuracy on the lifeQA dataset. That may because the CLIP matching is originally designed for image and caption matching, which is not robust to the image and question matching. Thanks to the adaptation of the text encoder through a learnable MLP block, our question guided retrieval can better identify the relevant video chunks.

**Effectiveness of Chunk-wise Design.** *Video-ChatGPT* performs video level spatial temporal pooling to obtain 356 vision tokens. Such global pooling would result in loss of details, especially when the question related video segments take a small portion of the entire video. In our design, we perform chunk level spatial temporal pooling to preserve more information about each chunk. Table 3 shows that when we uniform sample the chunks to have 340 vision tokens, *R-VLM w/Uni* obviously outperforms *Video-ChatGPT*, demonstrating the effectiveness of our chunk-wise design.

Table 4: Influence of soft matching (SM) loss, evaluated by the accuracy (%) / average score.

| Model | WildQA | QaEgo4D | lifeQA | Social-IQ 2.0 |
|---|---|---|---|---|
| R-VLM w/o SM | 59.94/3.28 | 31.12/2.36 | 36.29/2.47 | 57.22/3.17 |
| R-VLM | **64.82/3.39** | **32.51/2.43** | **38.71/2.61** | **63.65/3.40** |

**Influence of Soft Matching (SM) Loss.** In order to regularize the learning of the MLP block for better retrieval, we introduce SM loss. Table 4 shows the comparison of our framework without the SM loss (*R-VLM w/o SM*) and that with SM loss (*R-VLM*). We can see that incorporating the SM loss significantly improves the performance. Our final scheme with the SM loss *i.e.*, *R-VLM* outperforms that without SM Loss. SM Loss uses the cosine similarity between text embedding and video tokens as weight to re-weight the video tokens to obtain new video tokens. Then maximize the similarity between the new video tokens and the text embedding. This facilitates our learnable retrieval layer to find the video clips most similar to the questions.

**Influence of the Hyperparameter $K$.** In general, when $K$ is too small, it may lead to a loss of information necessary to answer the question. When $K$ is too large, interference may be introduced, confusing the LLM. We found $K$=5 presents a good trade-off on most datasets and set $K$ as 5. More details can be found in our Appendix. We leave the adaptive design of $K$ as the future work.

## 4.6 VISUALIZATION ANALYSIS

We visualize two examples from the QAEgo4D and WildQA datasets in Fig. 2 , with the following information. 1) The first row shows the video chunk samples by uniformly selecting 5 video chunks. 2) The second row shows the retrieved 5 chunks (ordered by time order) in our *R-VLM*. We mark the groudtruth chunks by red box. 3) We show the learned similarity score (see Eq. (2)) curve based on which the top $K$ chunks are selected. The horizontal axis represents the identity of chunk and the vertical axis denotes the similarity score of that chunk. The groundtruth chunks and our retrieved chunks are also marked. 4) The question and answers from different models: *R-VLM*, *R-VLM w/Uni.*, *Video-ChatGPT*, and *Video-LLaMA*, respectively.

For the QAEgo4D dataset, the average duration is about 8 minutes (120 video chunks), whereas the duration of the groundtruth segments is located 2% of the total video duration in average. It is difficult to hit groundtruth video segments by uniformly sampling $K(=5)$ chunks among 120 video chunks. In Fig. 2a, we can see that the fragments selected by uniform sampling are in general irrelevant to the problem, which is not conducive to the LLM reasoning. Our learnable retrieval can accurately find the segments where the answer lies in. Feeding the correct chunks to the LLM makes it possible to obtain the correct answer to the question.

For the WildQA dataset, groundtruth segments usually locate at different locations in the video and last for a period of time. Although uniform sampling sometimes hits a certain groundtruth segments, there is no guarantee. In contrast, our learned retrieval can correctly hit the segments where the groundtruth segments are located on. In this way, the LLM can better understand the video and answer some detailed questions (refer to "vegetation types" in Fig. 2b).

The predicted answers of our *R-VLM* is more accurate than *Video-ChatGPT*, and *Video-LLaMA*. More visualization can be found in our Supplementary.

Besides the successful cases, we present some failed cases in our Appendix (see Fig.5). There are two main cases of failure. One is that the retrieval did not select the correct video chunks. The other is that the retrieval correctly identified the correct video chunks, but the answer was wrong. For the later cases, we think more powerful vision feature extractor and LLMs would alleviate the problem.

## 5 CONCLUSION

The comprehension of long videos using LLMs remains an under-explored area. There are two main challenges associated with comprehending long videos. 1) Long videos generally lead to abundant vision tokens, which increase computational cost for LLM inference. 2) Global aggregation of vision tokens inevitably results in the loss of vision details especially when the question relevant video chunks take only a small portion of the entire video. Moreover, question irrelevant chunks introduce interference. In this work, we address these issues by introducing a simple yet effective retrieval-

based video language model (R-VLM) for long-video QA. Specifically, given a question (query) and a long video, our model identifies and selects the most question-relevant $K$ video chunks and uses their associated visual tokens to serve as context for the LLM inference. This effectively reduces the number of video tokens, preserves the most informative information, eliminates noise interference, and thus enhances system performance. Our experimental results demonstrate the effectiveness of our designs for comprehending long videos.

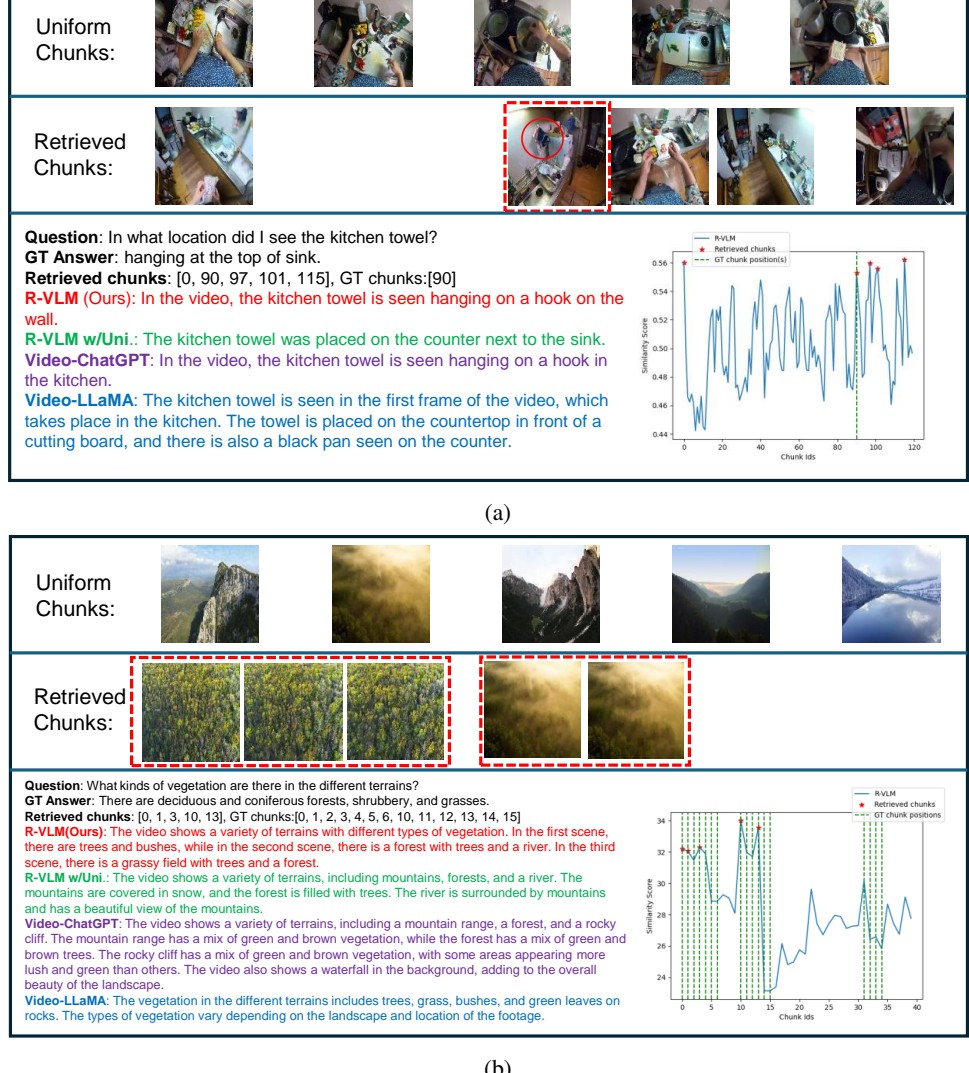

(a)

(b)

Figure 2: Visualization of video QA examples from (a) QAEgo4D and (b) WildQA. (a)The kitchen towel in question does not appear in the uniformly sampled video chunks. The second chunks selected by our model contain kitchen towel. Our answer states that the tower is hunging on a hook on the wall. Video-LLaMA answers incorrectly, where the towel does not appear in the first frame of the video, and it is not be placed on the countertop in front of a cutting board. (b)In this video, two clips show vegetation and the remaining clips show mountains, rivers, *etc*. Uniform sampling mainly obtains segments such as mountains and rivers rather than segments with vegetation. Therefore, only the terrain was answered, without giving vegetation types. In contrast, our retrieved chunks contain video clips of vegetation. Thus the types of vegetation are predicted correctly: trees, bushes, forest. Video-ChatGPT gives a global description and does not answer specific vegetation types.

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

# *Appendix*

## A    ABLATION STUDY ON INFLUENCE OF THE HYPERPARAMETER $K$

We study the influence of $K$. In general, when $K$ is too small, it may lead to a loss of information necessary to answer the question. When $K$ is too large, interference may be introduced, confusing the LLM. Table 5 shows the results of using different $K$. We found that as $K$ gradually increases from 1 to 5, the average performance increases. When $K$ increases from 5 to 7, the performance decreases. We found $K$=5 presents a good trade-off on most datasets, even though there are slight differences on different datasets. We leave the adaptive design of $K$ as the future work.

Table 5: Ablation study on the influence of $K$, evaluated in terms of accuracy (%)/score. We use bold to mark the best performance and underline to mark the second-best performance.

| Dataset | Video-ChatGPT | Ours(K=1) | Ours(K=3) | Ours(K=5) | Ours(K=7) |
|---|---|---|---|---|---|
| WildQA | 58.00/3.30 | 57.45/3.18 | 60.58/3.31 | **64.82/3.39** | 63.44/**3.39** |
| QaEgo4D | 29.74/2.43 | 32.42/2.41 | 32.04/2.42 | 32.51/**2.45** | **32.81**/2.42 |
| lifeQA | 33.87/2.55 | 37.63/2.62 | 38.44/2.62 | **38.71**/2.61 | 37.63/**2.65** |
| Social-IQ 2.0 | 57.73/3.26 | **63.92/3.44** | 60.89/3.34 | 63.65/3.40 | 62.89/3.34 |
| Average | 44.84/2.89 | 47.86/2.91 | 47.99/2.92 | **49.92/2.96** | 49.19/2.95 |

## B    MORE VISUALIZATION RESULTS

We visualize more examples from the QAEgo4D and WildQA datasets in Fig. 3 and Fig. 4, with the following information. 1) The first row shows the video chunk samples by uniformly selecting 5 video chunks. 2) The second row shows the retrieved 5 chunks (ordered by time order) in our *R-VLM*. We mark the groudtruth chunks by red box. 3) We show the learned similarity score curve based on which the top $K$ chunks are selected. The horizontal axis represents the identity of chunk and the vertical axis denotes the similarity score of that chunk. The groundtruth chunks and our retrieved chunks are also marked. 4) The question and answers from different models: *R-VLM*, *R-VLM w/Uni.*, *Video-ChatGPT*, and *Video-LLaMA*, respectively.

We also show some failure examples in Fig. 5. A detailed analysis of the reasons for failure is given in the figure caption. There are two main cases of failure. One is that the retrieval does not select the correct video chunks. The other is that the retrieval correctly identified the correct video chunks, but the answer is wrong. For the later cases, we think more powerful vision feature extractor and LLMs would alleviate the problem.

## C    COMPUTATIONAL COMPLEXITY

The computational cost comes from two parts. The first part is to encode the video frames through the CLIP encoder and the spatial-temporal pooling to get chunks. The second part is the retrieval of K=5 chunks and put them to LLM for inference. The spatial-temporal pooing and retrieval is very fast and negligible. On a single A100, we tested 120 60s videos from Social-IQ 2.0 and calculated the average inference time cost for a video. For a single video, the first part for vision feature extraction takes an average of 0.14s (in parallel for 60 frames), and the second part takes an average of 2.42s. The total time is 2.56s. Actually, for an even longer video, the time consumption for the second part does not increase since the input number of vision tokens is fixed (i.e., $68 \times 5 = 340$) in our scheme, which is favored for long video or streaming video understanding. The GPU memory consumption is about 17GB. Note that the computational cost for the LLM is proportional to the number of tokens.

The FLOPs for LLM inference can be roughly estimated as 2PD, where P denotes the number of parameters (model size), and D denotes the number of tokens. The computational complexity of LLM is proportional to the number of tokens which consists of text tokens (question and answer)

and vision tokens. The LLM model size P is 6.7B. On the training dataset, the average number of tokens for question and answers is 80, i.e., $D_{tex} = 80$. This varies on different datasets. For simplicity, we assume the number is the same for all the datasets. We denote the number of vision tokens as $D_{vis}$. The total number of tokens is $D = D_{tex} + D_{vis}$. For the four video datasets, WildQA, QaEgo4D, lifeQA, Social-IQ 2.0, the average number of vision chunks is 19, 122, 20, and 16, where each chunk has 68 tokens. Thanks to the retrieval, only $K = 5$ chunks ($D'_{vis} = 5 \times 68 = 340$ tokens) instead of all the chunks are needed as the input to LLM. Therefore, the computational cost (FLOPs) for LLM inference can be saved approximately $\frac{D_{vis} - D'_{vis}}{D_{tex} + D_{vis}}$, which are 69% (i.e., $(19 \times 68 - 5 \times 68)/(80 + 19 \times 68)$), 95%, 71%, and 64%, respectively.

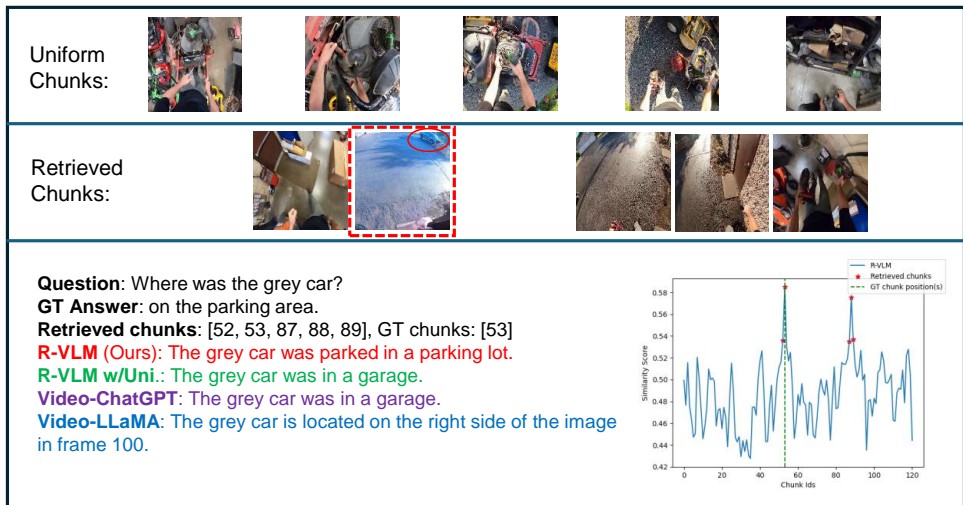

(a) We can see that the gray car does not appear in the uniformly sampled video chunks. Our R-VLM correctly answers that the car was parked in the parking lot (outdoors), but R-VLM w/Uni.'s answer was the garage (indoors). Video-LLaMA does not answer where the car is and the groundtruth frames do not appear in the frame 100. Video-ChatGPT made the similar mistake as R-VLM w/Uni.

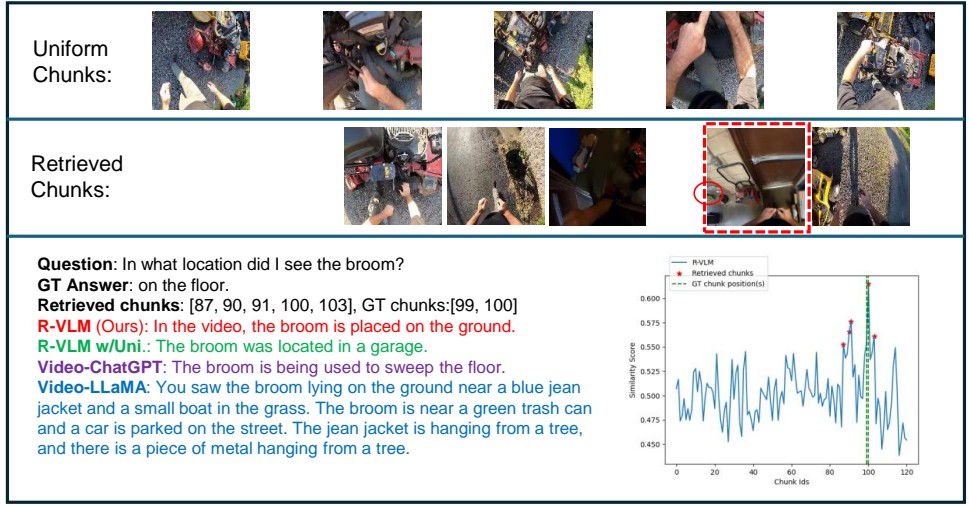

(b) The broom is small and is on the left in the red boxed image. Our R-VLM captures exactly where the broom is, i.e., on the ground. R-VLM w/Uni. does not capture the video chunks with broom and thus does not answer accurately. The answer of Video-ChatGPT is irrelevant to the question. The answer from Video-LLaMA is redundancy and tedious, where the mentioned blue jean jacket and boat actually do not appear in the video.

Figure 3: Visualization of video QA examples from QAEgo4D.

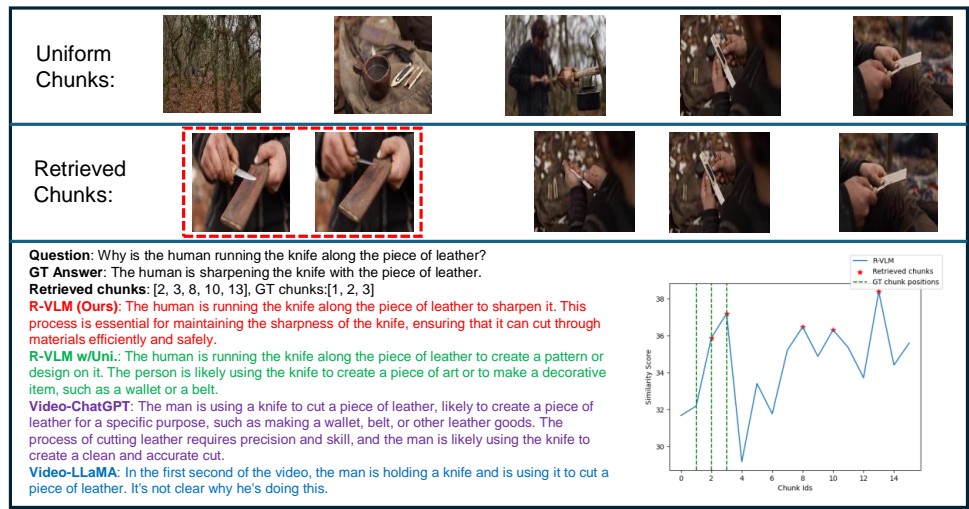

(a) The uniform sampling miss the chunks for sharpening process in GT-segs (at the beginning of video). As a result, LLM does not see the knife running along the leather, and only see the knife and some delicate small objects. Therefore, R-VLM w/Uni. mistakenly thought that this individual was carving patterns or making designs. Our retrieved chunks retain the process of the knife running on the leather and therefore R-VLM gives the correct answer. Both Video-LLaMA and Video-ChatGPT answered that people are cutting leather with a knife to make art, rather than sharpening the knife.

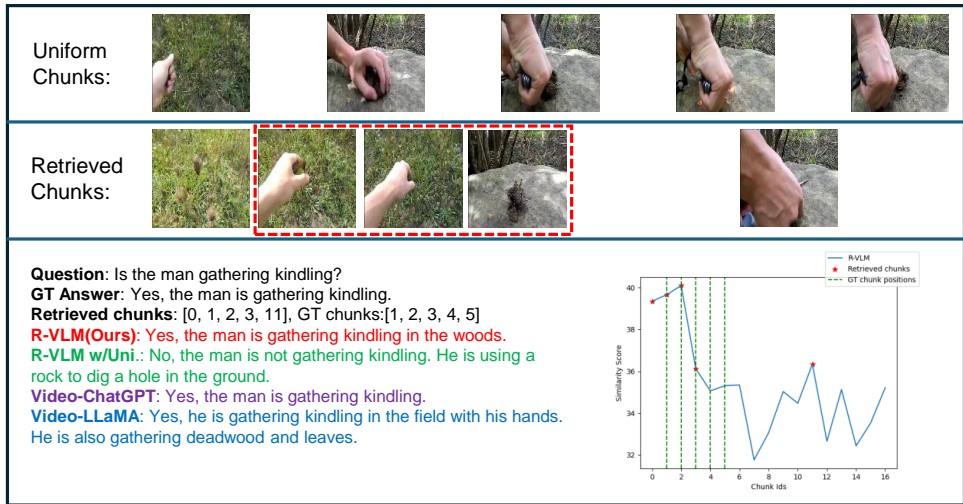

(b) In this video, collecting the kindling takes a short time, while placing the tinder on the stones takes a longer time. Uniform sampling makes LLM think that there is no process of collecting kindling and output the wrong answer of "digging a hole". Our R-VLM identified the relevant chunks of "gathering" even though those chunks only take a small duration in the entire video, generating correct answer. Video-LLaMA's prediction is not accurate since in fact the man did not gather deadwood and leaves in the video.

Figure 4: Visualization of video QA examples from WildQA.

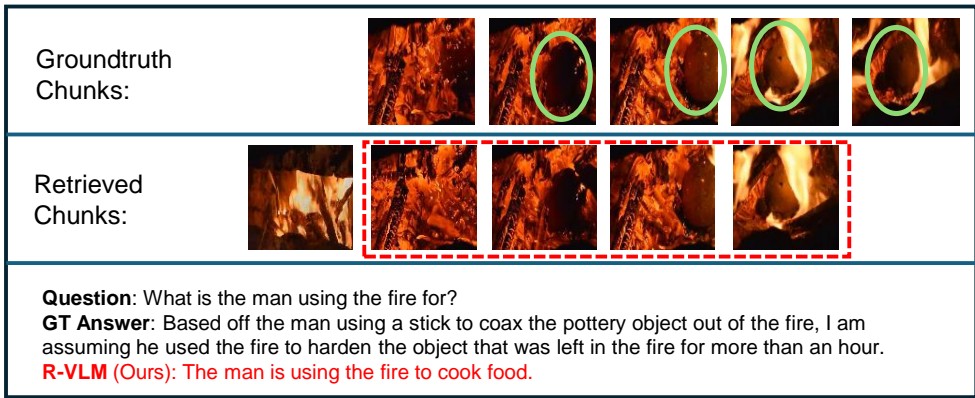

(a) A failure case from WildQA. This is a video of a person firing art. Although our method R-VLM retrieved the correct chunks, it gave the wrong answer of "cook food". We think this is due to the visual ambiguity of the target object and the biases of the LLM.

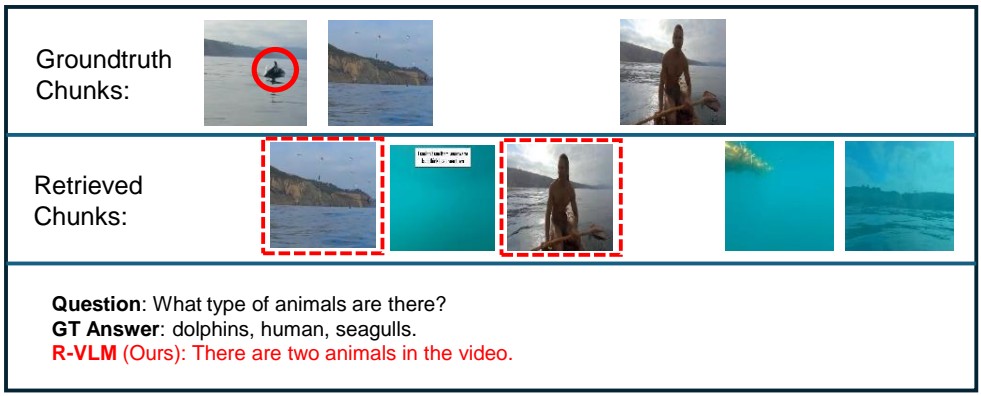

(b) A failure case from WildQA. Groundtruth chunks correspond to the chunks where three types of animals present, namely dolphins (the first chunk), seagulls, and human. Our method only retrieved the seagull and human chunks, but missed the dolphin chunk. R-VLM provided wrong answer due to the imperfect retrieval and the unsatisfactory reasoning capability of the used LLM.

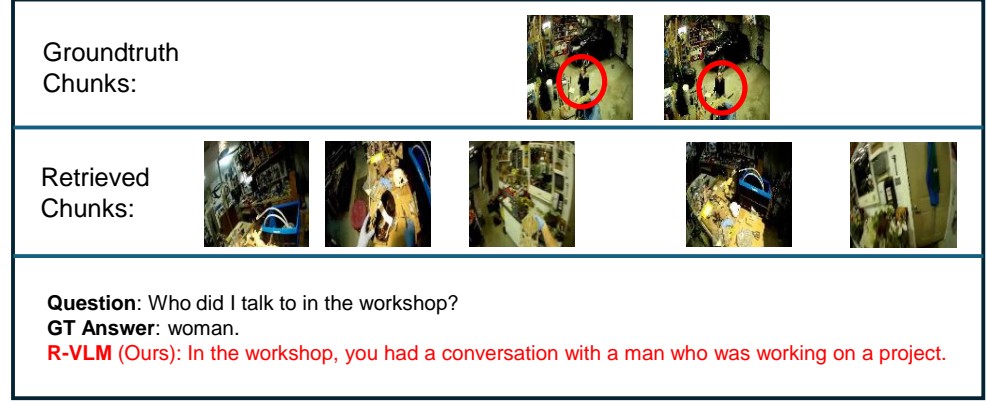

(c) A failure case from QAEgo4D. Our method did not find the correct chunks. Therefore, large language model did not correctly answer the question and provided hallucinated answer.

Figure 5: Visualization of failure cases from WildQA and QAEgo4D.

