# OpenReview forum: "Retrieval-Based Video Language Model for Efficient Long Video Question Answering"
_ICLR.cc/2024/Conference — Submitted to ICLR 2024_

### Official Review · Reviewer_qCRt · 2023-10-22

**Soundness:** 3 good
**Presentation:** 3 good
**Contribution:** 3 good
**Rating:** 5
**Confidence:** 4

**Summary:**

This paper addresses the challenges of comprehending long videos using LLMs by introducing a retrieval-based video language model (R-VLM) for long-video QA. The model identifies and selects the most question-relevant video chunks and their associated visual tokens to serve as context for LLM inference, effectively reducing the number of video tokens, preserving informative information, eliminating noise interference, and enhancing system performance.

**Strengths:**

1. The paper is well-written and easy to follow.
2. Using the visual tokens of the most question-relevant K video chunks as context for the LLM inference can effectively reduce computational costs.

**Weaknesses:**

1. Since there are not any ground-truth locations of the question relevant chunks for supervision, how to make sure that the learned chunk selection is effective?
2. In this paper, the authors set the number of video chucks to 5. How does the model perform with fewer or more video chucks? Will the performance of the model increase with more retrieved video chucks?
3. When evaluating the model on four video datasets, the authors should clarify how much computational cost is saved through the retrieval-based chuck selection mechanism.

**Questions:**

See Weaknesses.

**Details Of Ethics Concerns:**

In my opinion, no ethics review are needed.

---

> ### Author Response · Authors · 2023-11-18
> **Author Response to Reviewer qCRt**
>
> Thank you very much for your insightful review and constructive feedback. We have carefully considered each of your points and provided the detailed responses below.
>
> >1.Since there are not any ground-truth locations of the question relevant chunks for supervision, how to make sure that the learned chunk selection is effective?
>
> Response: This benefits from three aspects that promote the question (text) and vision alignment. First, the use of pretrained CLIP vision encoder and CLIP text encoder, where their features are trained to be matched/aligned. Second, the learnable MLP block (following the CLIP text encoder) makes it possible for the question and vision alignment, which the original CLIP pretraining (caption and vision alignment) has not been. Third, the proposed soft matching (SM) loss promotes the alignment of question and vision features. Meanwhile, the end-to-end instruction tuning (loss from the LLM) facilitates the selection of right chunks through the gradient back-propagation. Intuitively, only when the chunks are selected correctly, does the model generate the right answers. Table R4-1 shows the influence of these designs. We can see that they jointly make the retrieval-based solution feasible and efficient. R-VLM w/ CLIP M. denotes the use of the final CLIP class token feature of vision and text for matching instead of learnable retrieval. R-VLM w/o SM denotes the scheme without using SM loss. R-VLM w/o G. denotes our scheme but without back-propagating the gradient from the LLM loss. R-VLM denotes our final scheme.
>
> |  | **Video-ChatGPT** | **R-VLM w/ CLIP M.** | **R-VLM w/o SM** | **R-VLM w/o G.** |**R-VLM** |
> | --- | :---: | :---: | :---: | :---: | :---: |
> |  | **Acc(%)/Score** | **Acc(%)/Score** | **Acc(%)/Score** | **Acc(%)/Score** | **Acc(%)/Score** |
> | WildQA | 58.00/3.30 | 60.31/3.27 | 59.94/3.28 | 62.27/3.32 | **64.82/3.39** |
> | QaEgo4D | 29.74/2.43 | 31.52/2.43 | 31.12/2.36 | 31.66/**2.46** | **32.51**/2.45 |
> | lifeQA | 33.87/2.55 | 31.45/2.42 | 36.29/2.47 | 37.09/2.60 | **38.71/2.61** |
> | Social_IQ 2.0 | 57.73/3.26 | 61.17/3.28 | 57.22/3.17 | 62.43/3.36 | **63.65/3.40** |
>
> Table R4-1:  Effectiveness of our proposed retrieval module. R-VLM w/ CLIP M. denotes the use of the final CLIP class token feature of vision and text for matching instead of learnable retrieval. R-VLM w/o SM denotes the scheme without using SM loss. R-VLM w/o G. denotes our scheme but without back-propagating the gradient from the LLM loss. R-VLM denotes our final scheme.
>
> >2.In this paper, the authors set the number of video chucks to 5. How does the model perform with fewer or more video chucks? Will the performance of the model increase with more retrieved video chunks?
>
> Respond: Thank you for the question. Please see the ablation study from the global responses for all reviewers.
>
> >3.When evaluating the model on four video datasets, the authors should clarify how much computational cost is saved through the retrieval-based chuck selection mechanism.
>
> Respond: Thank you for the suggestion. The FLOPs for LLM inference can be roughly estimated as $2PD$, where $P$ denotes the number of parameters (model size), and $D$ denotes the number of tokens (see [here](https://medium.com/@dzmitrybahdanau/the-flops-calculus-of-language-model-training-3b19c1f025e4)). The computational complexity of LLM is proportional to the number of tokens which consists of text tokens (question and answer) and vision tokens. The LLM model size $P$ is 6.7B. On the training dataset, the average number of tokens for questions and answers is 80, i.e., $D_{tex} = 80$. This varies on different datasets. For simplicity, we assume the number is the same for all the datasets. We denote the number of vision tokens as $D_{vis}$. The total number of tokens is $D= D_{tex} + D_{vis}$.
> For the four video datasets, WildQA, QaEgo4D, lifeQA, Social-IQ 2.0, the average number of vision chunks is 19, 122, 20, and 16, where each chunk has 68 tokens. Thanks to the retrieval, only $K=5$ chunks ($D_{vis}' = 5 \times 68= 340$ tokens) instead of all the chunks are needed as the input to LLM. Therefore, the computational cost (FLOPs) for LLM inference can be saved approximately $\frac{D_{vis} – D_{vis}'}{D_{tex} + D_{vis}}$, which are 69\% (i.e., $(19\times68 – 5\times68)/(80 + 19\times68)$), 95\%, 71\%, and 64\%, respectively.
>
> Following your good suggestion, we have added the analysis to the updated appendix.

---

### Official Review · Reviewer_WHNQ · 2023-10-31

**Soundness:** 2 fair
**Presentation:** 3 good
**Contribution:** 2 fair
**Rating:** 5
**Confidence:** 4

**Summary:**

This paper proposes a Retrieval-based Video Language Model (R-VLM) for long video Question Answering (QA) tasks, aiming for efficient and interpretable long video QA. It computes the relevance between language token blocks and visual token blocks, selecting the top K visual token blocks as inputs for the Language Model (LLM) in VQA inference. This process matches and transmits the most relevant visual information to the LLM, reducing redundancy. Additionally, since the selected visual information is chosen based on its relevance to the question, this method also offers a degree of interpretability.

**Strengths:**

1. This method is relatively straightforward and easy to understand. By selecting the most relevant video segments, it reduces computational load, enhancing the efficiency and accuracy of the QA system.

2. Facing the growing content of long videos, this method provides a new perspective in the field of video understanding and retrieval.

**Weaknesses:**

1. Loss of detail: Although the method reduces distractions, key information might still be lost in the process of reducing visual tokens.

2. Lack of a dynamic adjustment mechanism: The model seems to lack the capability to dynamically adjust the number of selected video segments based on the content of the video, which could lead to information overload or oversimplification in some scenarios.

3. Resource consumption issues: Despite the reduction in the number of visual tokens, the overall resource consumption of the model (such as computational resources, memory, etc.) has not been fully discussed.

4. Regarding the selection of K, although the authors have explained it in the experimental section, the ablation of K still warrants exploration.

**Questions:**

1. How does the model handle the diversity and complexity of video content? Can it effectively process different types (such as documentaries, movies, news reports, etc.) and styles of videos?

2. When reducing visual tokens to improve efficiency, how is key information ensured not to be missed?

3. What is the model's generalizability? Whether it has good generalization capabilities for Out-Of-Distribution (OOD) data is worth discussing.

4. What about the real-time processing capability for long-duration videos? Is the model suitable for real-time video streams or scenarios requiring immediate responses?

5. In the ablation study section, the article does not conduct descriptive experiments on the hyperparameters used, such as $\lambda$ and K.

---

> ### Author Response · Authors · 2023-11-18
> **Author Response to Reviewer WHNQ(1/2)**
>
> Thank you very much for your insightful review and valuable suggestions/comments. We have carefully considered each of your points and provided the detailed responses below.
>
> >1.Loss of detail: Although the method reduces distractions, key information might still be lost in the process of reducing visual tokens. When reducing visual tokens to improve efficiency, how is key information ensured not to be missed?
>
> Response: Retrieval plays a very important role in reducing visual tokens, especially for long videos. For example, for a video of 180 seconds, we have 45 chunks in total. By retrieving the top 5 question-related chunks, the vision tokens are reduced by 89%. Targeting at dropping the question-irrelevant chunks, we try to ensure the key information is not missed. On the other hand, we found the spatial-temporal pooling to reduce the number of tokens in a chunk can still preserve the main information of a chunk, since the spatial-temporal redundancy in video is large. We did the ablation experiment of further compressing 256 spatial tokens to 64 spatial tokens in Video-ChatGPT and found that the performance only differs slightly (0.34%/0.27% on WildQA/lifeQA).
>
> How to make sure that the retrieving (selection of chunks) does not lose key information? The effectiveness of retrieval benefits from three aspects that promote the question (text) and vision alignment. First, the use of pretrained CLIP vision encoder and CLIP text encoder, where their features are trained to be matched/aligned. Second, the learnable MLP block (following the CLIP text encoder) makes it possible for the question and vision alignment, which the original CLIP pretraining (caption and vision alignment) has not been. Third, the proposed soft matching (SM) loss promotes the alignment of question and vision features. Meanwhile, the end-to-end instruction tuning facilitates the selection of right chunks through the gradient back-propagation. Intuitively, only when the chunks are selected correctly, does the model generate the right answers. Table R3-1 shows the influence of these designs. We can see that they jointly make the retrieval-based solution feasible and efficient. R-VLM w/ CLIP M. denotes the use of the final CLIP class token feature of vision and text for matching instead of learnable retrieval. R-VLM w/o SM denotes the scheme without using SM loss. R-VLM w/o G. denotes our scheme but without back-propagating the gradient from the LLM loss. R-VLM denotes our final scheme.
>
> |  | **Video-ChatGPT** | **R-VLM w/ CLIP M.** | **R-VLM w/o SM** | **R-VLM w/o G.** |**R-VLM** |
> | --- | :---: | :---: | :---: | :---: | :---: |
> |  | **Acc(%)/Score** | **Acc(%)/Score** | **Acc(%)/Score** | **Acc(%)/Score** | **Acc(%)/Score** |
> | WildQA | 58.00/3.30 | 60.31/3.27 | 59.94/3.28 | 62.27/3.32 | **64.82/3.39** |
> | QaEgo4D | 29.74/2.43 | 31.52/2.43 | 31.12/2.36 | 31.66/**2.46** | **32.51**/2.45 |
> | lifeQA | 33.87/2.55 | 31.45/2.42 | 36.29/2.47 | 37.09/2.60 | **38.71/2.61** |
> | Social_IQ 2.0 | 57.73/3.26 | 61.17/3.28 | 57.22/3.17 | 62.43/3.36 | **63.65/3.40** |
>
> Table R3-1:  Effectiveness of our proposed retrieval module. R-VLM w/ CLIP M. denotes the use of the final CLIP class token feature of vision and text for matching instead of learnable retrieval. R-VLM w/o SM denotes the scheme without using SM loss. R-VLM w/o G. denotes our scheme but without back-propagating the gradient from the LLM loss. R-VLM denotes our final scheme.
>
> >2.Lack of a dynamic adjustment mechanism: The model seems to lack the capability to dynamically adjust the number of selected video segments based on the content of the video, which could lead to information overload or oversimplification in some scenarios.
>
> Response: We also believe that a dynamic mechanism would result in superior results and would leave this as future work. For example, we could analyze the learned affinity scores from the retrieval and investigate whether some statistic characteristics are helpful to produce reasonable prediction. Fortunately, we find the hyper-parameter of K=5 presents good trade-off on most datasets. Maybe K=5 can avoid oversimplification while the overload is tolerable, where this has already reduced large redundancy/irrelevant content.

---

> ### Author Response · Authors · 2023-11-18
> **Author Response to Reviewer WHNQ(2/2)**
>
> >3.Resource consumption issues: Despite the reduction in the number of visual tokens, the overall resource consumption of the model (such as computational resources, memory, etc.) has not been fully discussed.
>
> Response: Thanks for the good suggestion. We added the discussion in the revised appendix. The computational cost comes from two parts. The first part is to encode the video frames through the CLIP encoder and the spatial-temporal pooling to get chunks. The second part is the retrieval of $K=5$ chunks and put them to LLM for inference. The spatial-temporal pooing and retrieval is very fast and negligible. On a single A100, we tested 120 60s videos from Social-IQ 2.0 and calculated the average inference time cost for a video. For a single video, the first part for vision feature extraction takes an average of 0.14s (in parallel for 60 frames), and the second part takes an average of 2.42s. The total time is 2.56s. Actually, for an even longer video, the time consumption for the second part does not increase since the input number of vision tokens is fixed (i.e., $68 \times 5 = 340$) in our scheme, which is favored for long video or streaming video understanding. The GPU memory consumption is about 17GB. Note that the computational cost for the LLM is proportional to the number of tokens (with more analysis in the updated appendix).
>
> >4.Regarding the selection of $K$, although the authors have explained it in the experimental section, the ablation of K still warrants exploration.
>
> Response: Thank you. Please see the ablation study from the global responses for all reviewers.
>
> >5.Hyperparameter λ.
>
> Response: We follow a widely used strategy to determine $\lambda$. Particularly, we observe the two loss terms at the early stage of training and then determine the value of $\lambda$ such that both loss terms are on the same order of magnitude. We set $\lambda=10$ in our experiments.
>
> >6.How does the model handle the diversity and complexity of video content? Can it effectively process different types (such as documentaries, movies, news reports, etc.) and styles of videos?
>
> Responds: Testing datasets contain diverse videos, ranging from movies, wild videos, first person videos, etc. For videos of different types and styles, our model presents consistent improvement over baseline models and previous methods. This benefits from the designed retrieval mechanism that can select the question-relevant video chunks for video understanding, avoiding the submergence of most informative vision information to the overall information, especially for long videos.
>
> The capability of dealing with diverse videos is also attributed to the generalizable LLM and CLIP vision encoder, which have already trained on abundant language data and diverse vision data. How to efficiently bridge them becomes one of the key factors to success.
>
> >7.What is the model's generalizability? Whether it has good generalization capabilities for Out-Of-Distribution (OOD) data is worth discussing.
>
> Response: The model has high generalizability. Our method outperforms baseline models and previous methods in the zero-shot setting tests (Table 2 and 3 in the manuscript), which contain diverse OOD data. The generalizability is also attributed to the generalizable LLM and CLIP vision encoder, which have already trained on abundant language data and diverse vision data.
>
> >8.What about the real-time processing capability for long-duration videos? Is the model suitable for real-time video streams or scenarios requiring immediate responses?
>
> Response: Our approach is capable of real-time inference. The computational cost comes from two parts. The first part is to encode the video frames through the CLIP encoder and the spatial-temporal pooling to get chunks. The second part is the retrieval of K=5 chunks and put them to LLM for inference. The spatial-temporal pooing and retrieval is very fast and negligible. We tested 120 60s videos from Social-IQ 2.0 and calculated the average inference time cost for a video. For a single video, the first part for vision feature extraction takes an average of 0.14s (in parallel for 60 frames), and the second part takes an average of 2.42s. The total time is 2.56s. Actually, for an even longer video, the time consumption for the second part does not increase since the input number of tokens is fixed (i.e., 68$\times$5=340).
>
> Hence our model is suitable for real-time video streams and scenarios tolerable for a delay of within 3 seconds to get the answers.

---

### Official Review · Reviewer_P6XZ · 2023-11-02

**Soundness:** 2 fair
**Presentation:** 3 good
**Contribution:** 1 poor
**Rating:** 6
**Confidence:** 4

**Summary:**

The authors propose a new approach called retrieval-based video language model (R-VLM) for long range video understanding. In R-VLM, the video is divided into 'N' chunks of 4 seconds and a retrieval module selects the best 'K' chunks for a given question. The selected chunks are passed as input to the frozen LLM for question answering. The authors then evaluate R-VLM on 4 long range video QA datasets such as WildQA, QaEgo4D, lifeQA and Social-IQ 2.0 datasets. Experimental results demonstrate that R-VLM strongly outperforms previous approaches.

**Strengths:**

**1. Clarity:** The authors did an excellent job in the presentation and clarity of writing. The methodology, experiments and results section are presented in a systematic way.

**2. Strong results on multiple datasets:** The proposed approach demonstrated strong results on multiple long range QA datasets.

**Weaknesses:**

**1. Insufficient Contributions vis-à-vis (Maaz et al.):** The contributions of this work are limited and doesn't meet ICLR standards. There are a only couple of differences between this work and Video-ChatGPT in terms of using chunks vs global pooling. To reduce the dimensionality of individual chunks, the authors adopt the same spatial-temporal pooling used in (Maaz et al.).

**2. Unclear advantages of chunks:** The main contributions in this work is dividing video into small chunks of 4 seconds. However, the authors use a very small sampling rate (1fps) which is essentially 4 frames for 4 seconds. Its unclear how chunks with small sampling rate provide an advantage to uniform/random sampling of equal no. of frames over the entire video while keeping the entire pipeline unchanged. There are currently no comparisons included in the paper.

**3. References:** There are some issues with the reference style used. The references got mixed up with the text and are sometimes indistinguishable.

**4. Relation to brain/biology:** Although the authors connect the proposed approach to brain, I believe such a comparison might be unnecessary. There are currently no clear established relations to deep learning and brain while most of them being assumptions.

**5. Lack of human studies in evaluation:** The authors use ChatGPT for evaluating the proposed model. ChatGPT is a LM trained on data which is not public and might have biases. Therefore, it is not an accurate evaluation method. The authors should add human evaluation by sampling 1000 questions/ground truth answers/model predictions and then calculating accuracy/average score for all the models.

**Questions:**

**1. More frames in a chunk vs less chunks:** Is there any advantage/dis-advantage of using more frames in a chunk and reducing the number of chunks?

**2. Results on QaEgo4D:** In Table-2, we observe that R-VLM significantly underperforms on QaEgo4D. What might be the reasons for this? Are there any significant difference between this dataset and others?

**3. Visualization results in Figure-2 (b):** The first three retrieved chunks and the last two are almost similar. However, the R-VLM is able to depict different terrains and vegetations. In fact, the uniform chunks have more diverse terrains than R-VLM.

---

> ### Author Response · Authors · 2023-11-18
> **Author Response to Reviewer P6XZ(1/2)**
>
> Thank you very much for your valuable suggestions/comments. Please see the detailed responses below.
>
> >1.Clarification of contributions. The main contribution in this work is dividing video into small chunks of 4 seconds. There are an only couple of differences between this work and Video-ChatGPT in terms of using chunks vs global pooling. To reduce the dimensionality of individual chunks, the authors adopt the same spatial-temporal pooling used in (Maaz et al.).
>
> Response: Thanks for the review. There are some misunderstandings regarding the main contributions of this work. We propose a retrieval-based video language model for efficient long video understanding. To the best of our knowledge, we are the first to validate the feasibility and efficiency of using retrieval for long video question answering with large language model. For a long video, the presence of abundant question-irrelevant content introduces interference to the video QA process, and the large amount of vision tokens increase computational cost for LLM inference. Therefore, we leverage question-guided retrieval to select relevant chunks for better video understanding. In order to compactly represent a chunk, we adopt spatial-temporal pooling as used in Video-ChatGPT to have chunk tokens and this is not claimed as our contribution. As pinpoint by Reviewer WHNQ, "Facing the growing content of long videos, this method provides a new perspective in the field of video understanding and retrieval."
>
> >2.Unclear advantages of chunks: The main contribution in this work is dividing video into small chunks of 4 seconds. However, the authors use a very small sampling rate (1fps) which is essentially 4 frames for 4 seconds. It's unclear how chunks with small sampling rate provide an advantage to uniform/random sampling of equal no. of frames over the entire video while keeping the entire pipeline unchanged. There are currently no comparisons included in the paper.
>
> Response: Thanks for the suggestion to build another baseline method VLM w/ vUni (v is short for varied). Following the suggestion, we uniform sample the equal no. of frames (i.e., $4 \times K = 20$ frames) over the entire video and then have $K=5$ chunks for fair comparison. For videos of different durations/lengths, this leads to different sampling rates and different time spans for chunks. When compared with our baseline scheme R-VLM w/ Uni which uniformly samples $K$ chunks, we do not expect which scheme should be better. They both lack the capability of excluding question-irrelevant content for robust video question answering. The results for R-VLM w/ Uni, VLM w/ vUni, and VLM (final) are shown in Table R2-1. We can see that VLM w/ vUni achieves better performance than our baseline method VLM w/ Uni while our retrieval-based solution R-VLM achieves the best performance.
>
> The current design of our chunks aims to facilitate retrieval, which makes each chunk of different videos at the same granularity (4 seconds) and may ease the learning of retrieval. Moreover, such chunk sampling manner is suitable for streaming video, where the provided chunks are at the same granularity and would not sacrifice the granularity (i.e., using longer duration for a chunk at coarse granularity) as the increase of video length. Correspondingly, we address the challenge from increased number of chunks by retrieving the most question-relevant chunks.
>
> |  | **Video-ChatGPT** | **R-VLM w/ Uni** | **R-VLM w/ vUni** | **R-VLM** |
> | --- | :---: | :---: | :---: | :---: |
> |  | **Acc(%)/Score** | **Acc(%)/Score** | **Acc(%)/Score** | **Acc(%)/Score** |
> | WildQA | 58.00/3.30 | 61.23/3.36 | 62.67/3.38 | **64.82/3.39** |
> | lifeQA | 33.87/2.55 | 36.56/2.56 | 36.82/2.66 | **38.71/2.61** |
>
> Table R2-1: Performance comparison of different uniform sampling methods and our method R-VLM.
>
> >3.Issues with the references style used.
>
> Response: Thank you! We have revised them in the updated manuscript.
>
> >4.Relation to brain/biology.
>
> Response: Thank you for the comment. We concur with your view that there are currently no clear established relations to deep learning and brain while most of being assumption. Our intention is to draw upon the concept of the retrieval mechanism in the brain as an analogy to aid readers in rapidly grasping the value of retrieval in comprehending lengthy videos. This concept also served as the inspiration behind our design.

---

> ### Author Response · Authors · 2023-11-18
> **Author Response to Reviewer P6XZ(2/2)**
>
> >5.Lack of human studies in evaluation.
>
> Response: We will incorporate more human evaluation. Currently, due to time limitations, we have conducted human studies in 100 samples of four test datasets on three models, Video-ChatGPT, R-VLM w/ Uni, and our scheme R-VLM. We have asked three people to blindly rate correct/wrong and scores and show the average evaluation results in Table R2-2. We can see that our method R-VLM achieves the best accuracy and score under human evaluation. The main trends from ChatGPT evaluation are consistent with our human evaluation, which also indicates ChatGPT evaluation has a certain degree of reliability and can be referenced even though imperfect. Currently, using 100 samples is still too small and we will enlarge the sample numbers to make them more reliable.
>
> | Evaluator | **Video-ChatGPT** | **R-VLM w/ Uni** | **R-VLM (Ours)** |
> | --- | :---: | :---: | :---: |
> |  | **Acc(%)/Score** | **Acc(%)/Score** | **Acc(%)/Score** |
> | ChatGPT | 48.00/2.98 | 44.00/3.02 | **50.00/3.03** |
> | Human | 42.33/2.26 | 37.67/2.09 | **46.33/2.41** |
>
> Table R2-2: The results of human evaluation and ChatGPT evaluation on 100 samples, which are randomly selected from four test datasets.
>
> >6.More frames in a chunk vs less chunks: Is there any advantage/dis-advantage of using more frames in a chunk and reducing the number of chunks?
>
> Response: In considering the memory limitation and video redundancy, we sample at a low frame rate (1fps). To allow a chunk to contain dynamic temporal information, we use 4 seconds that are expected to contain temporal dynamics as the chunk unit. Following the suggestion, we use more frames, i.e., 8 frames in a chunk at the same sample rate of 1fps. Table R2-3 shows the performance comparison. We can see that the performance of using 4 frames/chunk is better than that using 8 frames/chunk. That may be because using 4 frames/chunk allows a finer granularity retrieval. Both the two settings outperform the baseline method Video-ChatGPT.
>
> |  | **Video-ChatGPT**| **R-VLM(8frames/chunk)** | **R-VLM(4frames/chunk)** |
> | --- | :---: | :---: | :---: |
> |  | **Acc(%)/Score** | **Acc(%)/Score** | **Acc(%)/Score** |
> | WildQA | 58.00/3.30 | 62.58/3.37 | **64.82/3.39** |
> | QaEgo4D | 29.74/2.43 | 32.36/2.43 | **32.51/2.45** |
> | lifeQA | 33.87/2.55 | 36.83/**2.63** | **38.71**/2.61 |
> | Social_IQ 2.0 | 57.73/3.26 | 62.20/3.33 | **63.65/3.40** |
>
> Table R2-3: Table R2-3: Performance comparison of R-VLM (4frames/chunk) and R-VLM (8frames/chunk).
>
> >7.Results on QaEgo4D: In Table-2, we observe that R-VLM significantly underperforms on QaEgo4D. What might be the reasons for this? Are there any significant differences between this dataset and others?
>
> Response: R-VLM underperforms Video-LLaMa in terms of accuracy on QaEgo4D while **outperforms** Video-LLaMa significantly in terms of average score (2.45 vs. 1.94). We found Video-LLaMA is prone to give detailed description of the entire video, whereas the answer is usually question-irrelevant. Fig.2a in the manuscript and Fig.3b in the appendix show some typical examples on the QaEgo4D dataset, where the answer includes much information while the desired answer is submerged in the overall answer. In contrast, our model provides more concise and accurate answers.
>
> The accuracy metric from ChatGPT is sometimes not perfect. Being video-agnostic, ChatGPT may consider the answer "You saw the broom lying on the ground near a blue jean jacket and a small boat in the grass…" as correct even though "a blue jean jacket and a small boat" never appeared in the video (See Fig.3b in the appendix). Given the groundtruth answer "on the floor", the score of our model's answer "In the video, the broom is placed on the ground" is higher than that of Video-LLaMA. The joint consideration of accuracy and score may give more reliable evaluation.
>
> >8.Visualization results in Figure-2b: The first three retrieved chunks and the last two are almost similar. However, the R-VLM is able to depict different terrains and vegetations. In fact, the uniform chunks have more diverse terrains than R-VLM.
>
> Response: In the visualization of Fig2b, the question is "What kinds of vegetation are there in the different terrains?", which intends to know the type of vegetation, rather than to depict the various terrains. Our R-VLM retrieves all vegetation-related chunks. Although they are similar, those chunks are highly relevant to the question and helpful for answering it. Even though uniform chunks hit more diverse terrains, only the second chunk contains identifiable vegetation. Therefore, the scheme with uniform chunks(R-VLM w/Uni) generates **a wrong answer** that is irrelevant to the question.

---

> > ### Comment · Reviewer_P6XZ · 2023-11-20
> > **Re: Author response**
> >
> > I thank the authors for their effort in addressing my comments. I have read the rebuttal and the authors were able to address most of my concerns. They were also able to provide new evaluations which are crucial to understanding the effectiveness of the proposed approach. Therefore, I am raising my score.

---

> > > ### Author Response · Authors · 2023-11-20
> > > **Response to Reviewer P6XZ**
> > >
> > > We sincerely appreciate your positive feedback and the time you've taken to carefully reviewing our responses. Your valuable suggestions and comments are very important in improving the quality of our work.  Should you have any additional insights or inquiries, we warmly welcome them.

---

### Official Review · Reviewer_c7Li · 2023-11-03

**Soundness:** 2 fair
**Presentation:** 2 fair
**Contribution:** 2 fair
**Rating:** 3
**Confidence:** 3

**Summary:**

This work proposes a framework that retrieves the most relevant video chunks for a given question and projects their representations onto a pretrained Large Language Model (LLM): Vicuna. The LLM then uses these visual features in combination with question features to perform long video question-answering tasks. The experiments across four long video QA benchmarks show that the proposed framework outperforms two baseline models. Ablation studies reveal that the framework is more effective than methods solely based on temporal-spatial pooling, uniform sampling, or pretrained CLIP retrieval. Additionally, the importance of the soft matching loss is highlighted in the final ablation study.

**Strengths:**

+) The motivation for the proposed framework is clearly and effectively presented in the Introduction and Figure 1. The Introduction is well-crafted, making it easy to grasp the paper's concept.

+) The framework has been proven effective in capturing the key chunks relevant to the posed questions, outperforming the global temporal-spatial pooling baseline (Video-ChatGPT) and the Q-former baseline (Video-LLaMA) in four different long video QA benchmarks, despite Video-LLaMA being an off-the-shelf pretrained model not trained on the same data.

+) The ablation studies further confirm the effectiveness of the design decisions made in the framework. The learnable retrieval-based approach surpasses methods based on uniform sampling and pretrained CLIP representation retrieval.

+) The qualitative results show the framework's ability to select important video snippets to answer the given questions accurately.

**Weaknesses:**

-) Section 3.1 is poorly written and difficult to understand. It is unclear why two different spatial average pooling layers are used to obtain the global spatial average pool M tokens. It's also not clear why it is necessary to specify the N tokens as h/2 x w/2 from the first spatial average pooling. A more detailed explanation or an additional model figure could help clarify this process.

-) The paper lacks specific details, such as which variant of the pretrained CLIP model is used. Given that there are many variants that significantly outperform the original, this is an important detail. Also, the reason for using only 4 frames at 1 FPS per chunk is not explained; is this due to memory limitations?

-) Another average pooling is used for chunk token retrieval, but the rationale behind this is not clear. Previous works have shown the effectiveness of using a Q-former or a few transformer layers to summarize visual tokens. This design choice needs clarification. Furthermore, it is surprising that a simple linear layer is used to project the visual tokens into the LLM's inputs.

-) There is no ablation study to justify the choice of the hyperparameter K (=5). One could imagine that decreasing K might lead to a loss of information necessary to answer the question, while increasing K could provide more information but might also introduce redundancy, potentially confusing the LLM.

-) While qualitative results are provided, there is no discussion or analysis of the model's failure modes.

**Questions:**

o) Can the authors present more qualitative results, including both successful and unsuccessful examples?

o) Could the authors include the missing details and clarify the points of confusion regarding the use of many average pooling layers at different places and the simple linear projection of the visual tokens?

o) Could the authors conduct experiments with a Q-former for spatial-temporal visual feature aggregation as an alternative to average pooling?

o) Could the authors consider using a Q-former for projecting visual tokens instead of a simple linear layer?

o) Can the authors analyze the impact of different values for the hyperparameter K?

o) Could the authors finetune the vision-language branch proposed in Video-LLaMA using the same Video Instruction Data collected by Maaz et al. (2023) as was used to train the proposed framework for a fair comparison?

o) Would the authors consider revising Section 3.1 for better clarity?

---

> ### Author Response · Authors · 2023-11-18
> **Author Response to Reviewer c7Li(1/3)**
>
> Thank you very much for your thorough review, constructive feedback and encouraging comments. We have carefully considered and responded to each of your points.
>
> >1.Section 3.1 is poorly written and difficult to understand. It is unclear why two different spatial average pooling layers are used to obtain the global spatial average pool $M$ tokens. It's also not clear why it is necessary to specify the $N$ tokens as $h/2 \times w/2$ from the first spatial average pooling. Could the authors include the missing details and clarify the points of confusion regarding the use of many average pooling layers at different places?
>
>
> Response: Thank you for this nice suggestion. We have rewritten this section for clarity. In short, the purpose of pooling here is to reduce the number of tokens. The first spatial pooling aims to reduce the spatial feature resolution, which is equivalent to that we take the CLIP features of reduced resolution as the extracted feature. The spatial-temporal pooling as in Video-ChatGPT(Maaz et al. (2023)) aims to further reduce the number of tokens of a chunk while preserving spatial-temporal features.
>
> Why is the former pooling (to have $M \times h/2 \times w/2$ tokens) needed? Using spatial-temporal pooling as in Video-ChatGPT can compress the number of tokens in a chunk from 1024 to 4+256=260. We believe that compared to temporal information, the 256 tokens of spatial information still have much redundancy. We did some experiments by further compressing 256 tokens by spatial pooling to 64 tokens in Video-ChatGPT(Maaz et al. (2023)). We found the accuracy only dropped slightly by 0.34\%/0.27\% on WildQA/lifeQA. Therefore, we use this simple spatial pooling strategy to reduce the number of tokens by 75\%.
>
> We have revised the description in Section 3.1 as below and updated our manuscript:
> The preliminary token number of a chunk is large, i.e, $M \times h \times w = 4\times 16 \times 16 = 1024$. This would result in a high computational burden and large memory requirement for the LLM even though we only select a few chunks as input to LLM. We found that reducing the spatial resolution by a factor of 4 leads to marginal difference in performance while this can significantly reduce the number of tokens by 75\%. Therefore, we perform spatial average pooling with stride 2 to have $M \times \bar{h} \times \bar{w} = 4\times 8 \times 8 = 256$ tokens per chunk, where $\bar{h}$=$h/2$ and $\bar{w}$=$w/2$. This is equivalent to that we take the CLIP features of reduced resolution as the extracted feature. How to further reduce the number of tokens while preserving spatial-temporal features of a chunk? Motivated by _Video-ChatGPT_ \cite{maaz2023video}, we perform spatial-temporal pooling on the token tensor. Particularly, global spatial average pooling for each frame is performed and thus we obtain $M$ tokens for the $M$ frames (e.g., 4 tokens). Temporal pooling for each spatial position is performed to have $N = \bar{h} \times \bar{w} = 8 \times 8 = 64$ tokens. We have $N+M=64+4=68$ tokens $F_i^j \in \mathbb{R}^{(N+M)\times D}$, with $D$ dimension for each token. Compared with the original 1024 in a chunk, the number of tokens has been reduced to 6.6%, which brings calculation efficiency.
>
> >2.It is surprising that a simple linear layer is used to project the visual tokens into the LLM's inputs.
>
> Response: Previous works such as LLaVA (Liu et al. (2023)) and Video-ChatGPT (Maaz et al. (2023)) also use a simple linear layer to project the visual tokens into the LLM’s input. In LLaVA, together with using the linear layer, its LLM model is finetuned for better alignment and performance. We follow Video-ChatGPT to reuse the finetuned LLM in LLaVA and only tune the linear layer for the vision tokens (LLM is fixed). In addition, we think since the CLIP vision encoder is pretrained to align with text, it becomes easier to align with the text space as the LLM’s inputs.
>
> >3.The paper lacks specific details, such as which variant of the pretrained CLIP model is used. Given that there are many variants that significantly outperform the original, this is an important detail. Also, the reason for using only 4 frames at 1 FPS per chunk is not explained; is this due to memory limitations?
>
> Response: We use the pretrained CLIP ViT-L/14 (Radford et al. 2021) as used in LLaVA and Video-ChatGPT. The CLIP ViT-L/14 pretrained checkpoint download link is: https://huggingface.co/openai/clip-vit-large-patch14.
>
> Yes. In considering the memory limitation and video redundancy, we sample at a low frame rate. To allow a chunk to contain dynamic temporal information, we use 4 seconds that are expected to contain temporal dynamics as the chunk unit. We have added clarification in the revised manuscript.

---

> ### Author Response · Authors · 2023-11-18
> **Author Response to Reviewer c7Li(2/3)**
>
> >4.Another average pooling is used for chunk token retrieval, but the rationale behind this is not clear. Previous works have shown the effectiveness of using a Q-former or a few transformer layers to summarize visual tokens. This design choice needs clarification.
>
> Response: Thank you for the good suggestion. To identify the question-related chunks, we measure the affinity between the text feature vector and the chunk representation. For simplicity, we use the average pooled feature from the 68 tokens of the chunk as the chunk representation for the text and chunk matching. First, using average pooled feature is widely used in tasks such as video retrieval and action recognition. It provides a suitable representation of a video clip. Second, this is parameter-free, avoiding the need for large memory in optimization especially when the number of chunks is large. Therefore, we use this simple strategy to identify the related chunks and pass the spatial and temporal tokens of each selected chunk to LLM.
>
> >5.There is no ablation study to justify the choice of the hyperparameter K (=5).
>
> Response: Thanks. Please see the global responses for all reviewers.
>
> >6.While qualitative results are provided, there is no discussion or analysis of the model's failure modes. Can the authors present more qualitative results, including both successful and unsuccessful examples?
>
> Response: Thanks. We have added more discussion to the revised manuscript. For more successful examples, please refer to Fig.3 and Fig.4 in appendix. Examples of failures have also been randomly sampled and added to the updated appendix in Fig.5. We provide analysis for each example in the caption of the figures. There are two main cases of failure. One is that the retrieval does not select the correct video chunks. The other is that the retrieval correctly identified the correct video chunks, but the answer is wrong. For the later cases, we think more powerful vision feature extractor and LLMs would alleviate the problem.
>
> >7.Could the authors conduct experiments with a Q-former for spatial-temporal visual feature aggregation as an alternative to average pooling?
>
> Response: Thanks for the suggestion. We have thought about the use of Q-former to replace the spatial-temporal visual feature pooling in a chunk to reduce the number of tokens. The challenges lie in the large memory cost in training, especially for long videos with abundant chunks. Take a video of 8 minutes for example. There are 120 chunks to be processed by Q-former respectively, where the feature responses, gradient tensors require abundant memory in training and make it unbearable. Therefore, we use the parameter-free pooling to obtain chunk tokens.
>
> As an alternative, we use a Q-former to globally aggregate all the chunk tokens as the input of LLM, where the input to Q-former is the $L_i \times (N+M)$ tokens, where $L_i$ denotes the number of chunks, $N+M=68$ denotes the reduced number of tokens (through spatial-temporal pooling) in a chunk. The output is 340 vision tokens. There is no retrieval in this scheme, and we dub it VLM-Qformer. Table R1-1 shows the comparison with our scheme. We can see that our R-VLM outperforms VLM-Qformer significantly, thanks to the retrieval-based design which can exclude the inference of irrelevant vision tokens.
>
> |  | **VLM-Qformer** | **R-VLM (Ours)** |
> | --- | :---: | :---: |
> |  | **Acc(%)/Score** | **Acc(%)/Score** |
> | WildQA | 58.59/3.16 | **64.82/3.39** |
> | QaEgo4D | 30.58/2.35 | **32.51/2.45** |
> | lifeQA | 35.48/2.56 | **38.71/2.61** |
> | Social_IQ 2.0 | 54.30/3.14 | **63.65/3.40** |
>
> Table R1-1: Performance comparison of our scheme and a baseline scheme VLM-Qformer that uses Q-former to globally aggregate all the chunk tokens as the input to LLM.

---

> ### Author Response · Authors · 2023-11-18
> **Author Response to Reviewer c7Li(3/3)**
>
> >8.Could the authors consider using a Q-former for projecting visual tokens instead of a simple linear layer?
>
> Response: For the $K(N+M)=340$ vision tokens of the selected top K chunks, similar to LLaVA and Video-ChatGPT, a linear layer is used to project each token to the LLM input space. Following the reviewer’s suggestion, we use a Q-former to project the vision tokens (input $K(N+M)=340$ vision tokens and output the same number of tokens) while keeping other designs unchanged. We found the performance decreases by 1% and 3.3% in QaEgo4D and lifeQA in accuracy, respectively. That may be because Q-former is difficult to train even though such a powerful subnet is expected to lead to better performance.
>
> >9.Could the authors finetune the vision-language branch proposed in Video-LLaMA using the same Video Instruction Data collected by Maaz et al.(2023) as was used to train the proposed framework for a fair comparison?
>
> Response: Thanks for the suggestion. We follow your suggestion and finetune Video-LLaMA using the same Video Instruction Data collected by Maaz et al. (2023) and show the results below. We have two observations. (1)The performance of our method R-VLM is higher than Video-LLaMA-tuned and Video-LLaMA, demonstrating the effectiveness of our scheme. (2) The score of Video-LLaMA-tuned is higher than Video-LLaMA, while the accuracy is lower than Video-LLaMA. We found Video-LLaMA without tuning is prone to give very detailed description, but the answer is usually question-irrelevant. Below we show a typical example, where the answer from Video-LLaMA includes much information while the desired answer is submerged in the overall answer. Human is prone to rate a low score for such cases. We found that ChatGPT evaluator judges such redundancy answer as correct while rates a low score (see below example).  In contrast, Video-LLaMA-tuned's answer is more concise and clearer than Video-LLaMA. Therefore, we think when comparing two models trained with different datasets, score is a more reliable metric for evaluation.
>
> An example from QaEgo4D:
> Question: What paint can did I open?
> Answer: black paint.
> Video-LLaMA-tuned: You opened a black paint can and wiped it on a piece of wood.(Evaluation score: 4)
> Video-LLaMA: In the first second of the video, you can see a man holding a paint can and a person is painting a kitchen table with brown and black paint. The can is red and has the word "paint" written on it. There is a white chair, black and silver chairs, and a black and red couch in the room.(Evaluation score: 3)
>
> |  | **Video-LLaMA** | **Video-LLaMA-tuned** | **R-VLM** |
> | --- | :---: | :---: | :---: |
> |  | **Acc(%)/Score** | **Acc(%)/Score** | **Acc(%)/Score** |
> | WildQA | 63.19/3.18 | 61.20/3.24 | **64.82/3.39** |
> | QaEgo4D | **35.35**/1.94 | 28.40/2.23 | 32.51/**2.45** |
> | lifeQA | 35.75/2.32 | 34.68/2.39 | **38.71/2.61** |
> | Social_IQ 2.0 | 55.78/2.90 | 52.18/3.05 | **63.65/3.40** |
>
> Table R1-2: Performance comparison of Video-LLaMA before and after tuned using Video Instruction Data collected by Maaz et al. (2023) , and our method R-VLM.

---

### Author Response · Authors · 2023-11-18
**Author Response to All Reviewers:**

Thank you very much for your positive feedback and constructive comments. We sincerely appreciate your recognition of the clear motivation behind our work, the efficacy of the method, the quality of the overall presentation, and the value of our perspective in long video understanding.

Your comments/suggestions are very helpful for the improvement of our paper. We have carefully considered them and provided detailed responses for each reviewer. Our manuscript has been updated accordingly (with the main changes highlighted by blue in the attached file). Should there be any further questions or concerns, we warmly invite you to share them. We also wish to extend our gratitude to the Area Chair for the diligent efforts in managing the review process of our paper.




>Ablation study on the choice of the hyperparameter K.

Thanks for the suggestion. Table 1 shows the results of ablation study on different K values. We found that as K gradually increases from 1 to 5, the average performance increases. When K increases from 5 to 7, the performance decreases. In general, when K is too small, it leads to a loss of information necessary to answer the question. When K is too large, interference may be introduced, confusing the LLM. We found K=5 presents a good trade-off on most datasets, even though there are slight differences on different datasets. We will leave the adaptive design of K as the future work.

|  | **Video-ChatGPT** | **Ours(K=1)** | **Ours(K=3)** | **Ours(K=5)** |**Ours(K=7)** |
| --- | :---: | :---: | :---: | :---: | :---: |
| WildQA | 58.00/3.30 | 57.45/3.18 | 60.58/3.31 | **64.82/3.39** | _63.44_/**3.39** |
| QaEgo4D | 29.74/_2.43_ | 32.42/2.41 | 32.04/2.42 | _32.51_/**2.45** | **32.81**/2.42 |
| lifeQA | 33.87/2.55 | 37.63/_2.62_ | _38.44_/_2.62_ | **38.71**/2.61 | 37.63/**2.65** |
| Social_IQ 2.0 | 57.73/3.26 | **63.92/3.44** | 60.89/3.34 | _63.65/3.40_ | 62.89/3.34 |
| Average | 44.84/2.89  | 47.86/2.91 | 47.99/2.92 | **49.92/2.96** | _49.19/2.95_ |

Table 1: Ablation study on the choice of the hyperparameter K, evaluated in terms of accuracy (\%)/score. We use bold to mark the best performance and italics to mark the second-best performance.

---

### Meta-Review · Area_Chair_9sK2 · 2023-12-11

**Metareview:**

This paper was reviewed by three experts and received mixed scores. Though all reviewers agree some aspects of the paper are promising, they also consistently raise concerns listed below.

1. The technical contribution of the proposed model is incremental (P6XZ).

2. The clarity of presentations needs to be improved (c7Li).

3. The experiments are limited. More ablation studies and comparisons are required to verify the model (c7Li, WHNQ).

While the research demonstrated indeed has promise, the decision is not to recommend acceptance in its current state. The authors are encouraged to consider the reviewers' comments when revising the paper for submission elsewhere.

**Justification For Why Not Higher Score:**

1. The technical contribution of the proposed model is incremental (P6XZ).

2. The clarity of presentations needs to be improved (c7Li).

3. The experiments are limited. More ablation studies and comparisons are required to verify the model (c7Li, WHNQ).

**Justification For Why Not Lower Score:**

NA

---

### Decision · Program_Chairs · 2024-01-16

Reject